# ß-arrestin 2 germline knockout does not attenuate opioid respiratory depression

**Iris Bachmutsky[1,2†], Xin Paul Wei[1,3†], Adelae Durand[1], Kevin Yackle[1*]**

[1]Department of Physiology, University of California, San Francisco, San Francisco, United States; [2]Neuroscience Graduate Program, University of California, San Francisco, San Francisco, United States; [3]Biomedical Sciences Graduate Program, University of California, San Francisco, San Francisco, United States

**Abstract** Opioids are perhaps the most effective analgesics in medicine. However, between 1999 and 2018, over 400,000 people in the United States died from opioid overdose. Excessive opioids make breathing lethally slow and shallow, a side-effect called opioid-induced respiratory depression. This doubled-edged sword has sparked the desire to develop novel therapeutics that provide opioid-like analgesia without depressing breathing. One such approach has been the design of so-called 'biased agonists' that signal through some, but not all pathways downstream of the μ-opioid receptor (MOR), the target of morphine and other opioid analgesics. This rationale stems from a study suggesting that MOR-induced ß-arrestin 2 dependent signaling is responsible for opioid respiratory depression, whereas adenylyl cyclase inhibition produces analgesia. To verify this important result that motivated the 'biased agonist' approach, we re-examined breathing in ß-arrestin 2-deficient mice and instead find no connection between ß-arrestin 2 and opioid respiratory depression. This result suggests that any attenuated effect of 'biased agonists' on breathing is through an as-yet defined mechanism.

**\*For correspondence:**
Kevin.Yackle@ucsf.edu

†These authors contributed equally to this work

**Competing interest:** The authors declare that no competing interests exist.

## Introduction

More than 180 people died each day in 2018 from depressed breathing following an opioid overdose, a lethal side-effect termed opioid induced respiratory depression (OIRD) (*Pattinson, 2008*; *Scholl et al., 2018*). Nevertheless, opioids remain among the most effective and widely prescribed analgesics, as evidenced by the World Health Organization's step ladder for pain management. These two contrasting characteristics have created the urgent desire to develop or discover novel μ-opioid receptor (MOR) agonists that provide analgesia without depressing breathing. Such a possibility emerged in 2005 when it was reported that mice with germline deletion of ß-arrestin 2 (*Arrb2-/-*) experience enhanced analgesia with attenuated respiratory depression when administered systemic morphine (*Raehal et al., 2005*). This finding inspired a new field of 'biased agonist' pharmacology with the goal of engaging MOR-dependent G-protein but not ß-arrestin 2 signaling. This approach remains a central strategy in drug discovery for analgesics (*Turnaturi et al., 2019*).

Germline deletion of the μ-opioid receptor gene (*Oprm1*) completely eliminates OIRD in murine models (*Dahan et al., 2001*) and selective deletion of *Oprm1* within the brainstem medullary breathing rhythm generator, the preBötzinger Complex (preBötC) (*Smith et al., 1991*), largely attenuates OIRD (*Bachmutsky et al., 2020*; *Varga et al., 2019*). When engaged, MOR G-protein signaling activates inwardly rectifying potassium channels (*Al-Hasani and Bruchas, 2011*) and inhibits synaptic vesicle release (*Zurawski et al., 2019*), thereby depressing neural signaling. Additionally, the MOR signals intracellularly via a pathway dependent on MOR internalization by ß-arrestin 2 (*Calebiro et al., 2010*; *Luttrell and Lefkowitz, 2002*). Indeed, all three pathways have been implicated in OIRD (*Montandon et al., 2016*; *Raehal et al., 2005*; *Wei and Ramirez, 2019*) but a role for *Arrb2* has been suggested to be important and selective for respiratory depression, relative to analgesia.

**eLife digest** Opioid drugs are commonly prescribed due to their powerful painkilling properties. However, when misused, these compounds can cause breathing to become dangerously slow and shallow: between 1999 and 2018, over 400,000 people died from opioid drug overdoses in the United States alone.

Exactly how the drugs affect breathing remains unclear. What is known is that opioids work by binding to specific receptors at the surface of cells, an event which has a ripple effect on many biochemical pathways. Amongst these, research published in 2005 identified the β-arrestin 2 pathway as being responsible for altering breathing. This spurred efforts to find opioid-like drugs that would not interfere with the pathway, retaining their ability relieve pain but without affecting breathing. However, new evidence is now shedding doubt on the conclusions of this study.

In response, Bachmutsky, Wei et al. attempted to replicate the original 2005 findings. Mice with carefully controlled genetic background were used, in which the genes for the β-arrestin 2 pathway were either present or absent. Both groups of animals had similar breathing patterns under normal conditions and after receiving an opioid drug. The results suggest β-arrestin 2 is not involved in opioid-induced breathing suppression.

These findings demonstrate that research to develop opioid-like drugs that do not affect the β-arrestin 2 pathway are based on a false premise. Precisely targeting a drug's molecular mechanisms to avoid suppressing breathing may still be a valid approach, but more research is needed to identify the right pathways.

Since the original OIRD study of *Arrb2-/-* mice, multiple MOR agonists, dubbed 'biased agonists', have been created that signal through G-protein but not ß-arrestin 2 pathways. These agonists are reported to produce analgesia with reduced respiratory depression (*Manglik et al., 2016*; *Schmid et al., 2017*) and thereby provide pharmacological support for the proposed role of ß-arrestin 2 in OIRD. However, recently, these studies have been called into question (*Hill et al., 2018*; *Kliewer et al., 2020*), prompting us to independently investigate the underlying necessity of ß-arrestin 2 in mediating ORID. Here, in an experiment where we rigorously control for genetic background, we demonstrate that basal breathing and OIRD are similar in *Arrb2+/+*, *Arrb2+/-*, and *Arrb2-/-* littermates. Furthermore, the in vitro preBötC rhythm is similarly silenced by an MOR agonist in all three genotypes. Our data, together with another recent report, does not show a role of *Arrb2* in opioid-induced respiratory depression and suggests that MOR biased agonists attenuate OIRD through a different mechanism.

## Results

Breathing behaviors and OIRD severity differ between strains of mice (*Bubier et al., 2020*). Therefore, we sought to compare basal and morphine depressed breathing in mice with the same genetic background. We bred F1 *Arrb2+/-* mice to generate littermates that were wildtype (+/+), heterozygous (+/-), and homozygous (-/-) for germline deletion of *Arrb2* (*Figure 1A*). Breathing was assessed by whole-body plethysmography following intraperitoneal (IP) injection of saline (for control recordings) or after IP morphine (20 mg/kg) one day later (*Figure 1B*). This same breathing protocol was conducted in air with 21% $O_2$ and 0% $CO_2$ (hereby referred as normoxic) and air with 21% $O_2$ and 5% $CO_2$ (hereby referred as hypercapnic). The hypercapnic state minimizes potential confounding fluctuations in breathing rate and depth seen in normoxic conditions. We used our previously described analytical pipeline (*Bachmutsky et al., 2020*) to assay the two key parameters that define OIRD, slow and shallow breathing. Slow breathing is measured as the instantaneous frequency of each breath and shallow breathing is defined by the peak inspiratory flow since it strongly correlates with the volume of air inspired (PIF, *Figure 1C*; *Bachmutsky et al., 2020*).

In the normoxic condition, the morphology of single breaths, respiratory rate, and peak inspiratory airflow after IP saline appeared similar in *Abbr2+/+* and *-/-* mice (*Figure 2A–B*, *Table 1* contains mean ± SEM and 95% CI). As expected for OIRD, IP morphine decreased the frequency and PIF, but the breathing characteristics remained indistinguishable in *Abbr2+/+* versus *-/-* mice (*Figure 2A–B*,

**Figure 1.** Experimental approach to measure OIRD in each *Arrb2* genotype. (**A**) Breeding scheme to generate F2 *Arrb2*+/+, +/-, and -/- littermates. (**B**), Whole body plethysmography experimental scheme. On Day 1, recordings were performed 15 min after IP saline injection. Day 2, recordings 15 min after IP morphine (20 mg/kg). Recordings were first conducted under normoxic conditions (21% $O_2$, 0% $CO_2$) and then at least one week later under hypercapnic conditions (21% $O_2$, 5% $CO_2$). (**C**), Example analysis of a single breath. The approximated airflow (mL/s) was used to identify inspiration (insp. <0 mL/s) and expiration (expir. >0 mL/s). Instantaneous frequency (Hz, $s^{-1}$) defined as the interval between inspiration onset and expiration offset. PIF, peak inspiratory airflow. These two parameters were used to define OIRD.

*Table 1*). Consistently, histograms of the instantaneous frequency and PIF for each breath after IP morphine showed overlapping distributions from *Arrb2*+/+ (combined from n = 5 mice), +/- (n = 6), and -/- (n = 7) animals (*Figure 2C,E*). We quantified OIRD as the ratio of the average instantaneous frequency or PIF after IP morphine normalized to IP saline. As expected from the raw data, OIRD was similar among the genotypes (rate decreased 60% and PIF by 40%, *Figure 2D and F*, *Table 2* contains 95% CI for each mean and the comparisons). In fact, there was a small attenuation of respiratory rate depression in *Arrb2*+/+ compared to *Arrb2*-/- and +/- mice, inconsistent with the hypothesis that

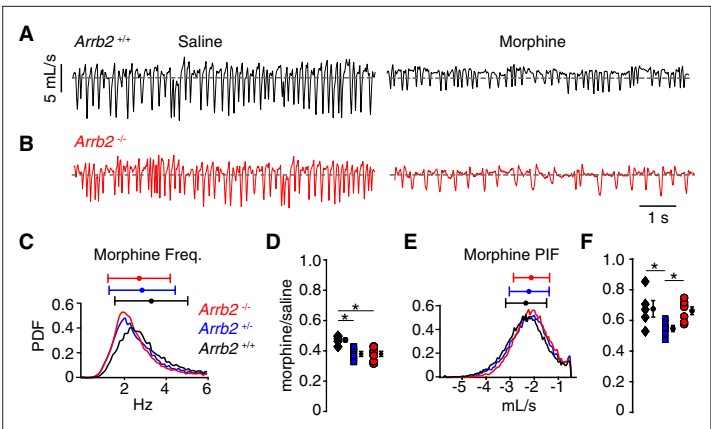

**Figure 2.** Basal respiration and OIRD in *Arrb2* littermates in normoxic conditions. (A) Example breathing trace in normoxic conditions (21% $O_2$, 0% $CO_2$) following IP saline (left) and morphine (right) for *Arrb2*+/+. (**B**), Example breathing traces from *Arrb2*-/-. (**C**), Histogram of instantaneous respiratory frequency (Hz) for all breaths in morphine from *Arrb2*-/- (red, combined from n = 7 animals), *Arrb2*+/- (blue, n = 6), and *Arrb2*+/+ (black, n = 5). PDF, probability density function. Top, mean (circle)± standard deviation (bars). Values of respiratory measurements are reported in Table 1. (**D**), OIRD defined as the ratio of average respiratory frequency in morphine to saline for *Arrb2*-/-, *Arrb2*+/-, and *Arrb2*+/+. Mean (circle)± standard deviation (bars). Data values included in Table 2. Note, *Arrb2*+/+ mice have less OIRD when compared to *Arrb2*+/- and -/-. (**E–F**), Analysis of peak inspiratory airflow (PIF) displayed as in **C–D**. Note, PIF in *Arrb2*±mice shows more OIRD when compared to *Arrb2*+/+ and -/-. There is no statistically significant difference between *Arrb2*+/+ and -/-. Single and Two Factor ANOVA and unpaired t-test statistics reported in Table 2. *, indicates the post-hoc single factor ANOVA comparisons with p-value < 0.05. Statistics were not corrected for multiple comparisons to maximize the possibility of identifying differences between *Arrb2* genotypes.

The online version of this article includes the following figure supplement(s) for figure 2:

**Source data 1.** Raw respiratory data, OIRD ratio, and statistical tests for recordings performed in normoxic conditions.

**Table 1.** Mean and confidence interval for normoxic condition raw respiratory frequency and peak inspiratory airflow after saline and morphine intraperitoneal injection.

| | Arrb2 -/- mean ± SEM | Arrb2 -/- 95% CI | Arrb2+/- mean ± SEM | Arrb2 +/- 95% CI | Arrb2 +/+ mean ± SEM | Arrb2 +/+ 95% CI |
|---|---|---|---|---|---|---|
| Freq. saline (Hz) | 7.05 ± 0.34 | 6.38 → 7.72 | 7.51 ± 0.41 | 6.71 → 8.31 | 6.93 ± 0.32 | 6.30 → 7.56 |
| Freq. morphine | 2.67 ± 0.16 | 2.36 → 2.98 | 2.83 ± 0.14 | 2.56 → 3.10 | 3.27 ± 0.16 | 2.96 → 3.58 |
| PIF saline (mL/s) | –3.17 ± 0.15 | –2.88 → –3.46 | –4.03 ± 0.25 | –3.54 → –4.52 | –3.5 ± 0.23 | –3.04 → –3.95 |
| PIF morphine | –2.09 ± 0.09 | –1.91 → –2.23 | –2.19 ± 0.10 | –1.99 → –2.39 | –2.33 ± 0.10 | –2.13 → –2.53 |

*Arrb2* mutation attenuates OIRD. These data lead us to conclude that OIRD in normoxic conditions is not diminished in *Arrb2-/-* mice.

These same breathing assays and analysis were also performed in a hypercapnic state. Hypercapnia eliminates any changes in breathing rate and depth that are simply due to variation in behavioral state, like sniffing versus calm sitting. Thus, although breathing when hypercapnic is faster and deeper (like in *Figure 3A*, *Table 3*), the reduced variability minimizes any chance that the conclusions in the normoxic condition are due to additional effects of opioids on behaviors such as sedation and locomotion, or non-opioid-related differences in arousal. Importantly, OIRD is still robustly observed in the hypercapnic state (*Figure 3A*, *Table 4*). As anticipated, the opioid depression of instantaneous frequency (by ~30%) and PIF (by ~30%) were the same among all three genotypes (*Figure 3C–F*, *Table 4*). Therefore, OIRD is not diminished in *Arrb2*-deficient mice in two independent breathing assays.

To understand the confidence of these results, we determined the extent that *Arrb2+/+* breathing parameters must be depressed (compared to *Arrb2-/-*) in order to produce a significant test statistic more than 80 % of the time, that is, power analysis. Given our cohort sizes and the observed variation in breathing parameters from *Arrb2+/+* and -/- littermates, we plotted the relationship between power (0–1) and percent difference between the *Arrb2-/-* and hypothetical *Arrb2+/+* means (see methods). When comparing the *Arrb2-/-* measured and *Arrb2+/+* hypothetical means, we could confidently distinguish these mean breathing frequencies so long as they differed by at least ~12–20%, and mean PIFs by more than ~10%–22% (*Figure 3—figure supplement 1*). Thus, our experimental approach enabled us to only detect mild to large differences in OIRD between *Arrb2+/+* and -/- littermates, if they had occurred. It remains possible that a small effect, much smaller than previously reported, was not identified in our study.

The most important site for OIRD is the preBötC (*Bachmutsky et al., 2020*). So, we directly measured the effects of the opioid peptide [D-Ala, *N*-MePhe, Gly-ol]-enkephalin (DAMGO) on the preBötC rhythm in all three *Arrb2* genotypes. Importantly, DAMGO robustly stimulates both MOR signaling pathways, G-protein and ß-arrestin 2 dependent signaling. The preBötC slice was prepared from *Arrb2* littermates and the genotype (+/+, +/-, -/-) was determined post hoc. The rhythm was monitored by measuring electrical activity in the hypoglossal nerve rootlet (*Smith et al., 1991*) for 20 minutes at baseline, and then with 20 nM then 50 nM DAMGO (*Figure 4A–B*). The preBötC rhythms of *Arrb2+/+*, +/-, and -/- littermates (n = 5, 20, 6) similarly slowed by ~70–80% at 20 nM DAMGO and nearly all were silenced at 50 nM (*Figure 4C–D*). If anything, it appears the *Arrb2-/-* slices showed a statistically significant increase in DAMGO sensitivity when compared to *Arrb2+/-* littermates

**Table 2.** OIRD values of respiratory frequency and peak inspiratory airflow in normoxic conditions and the several types of statistical tests.

The respiratory frequency OIRD in *Arrb2+/+* is larger than *Arrb2+/-* and -/-. The PIF OIRD for *Arrb2+/-* is smaller than *Arrb2+/+* and -/-.

| | Arrb2 -/- OIRD mean (95% CI) | Arrb2+/- mean (95% CI) | Arrb2 +/+ mean (95% CI) | Arrb2 -/- vs. Arrb2 +/+ (t-test) | Arrb2 -/- vs. Arrb2 +/+ (t-test 95% CI) | One-way anova | Tukey HSD/ Kramer Arrb2 -/- vs. +/+ | Two-way anova regression (interaction) |
|---|---|---|---|---|---|---|---|---|
| Freq. | 0.38 (0.35→0.41) | 0.38 (0.35→0.41) | 0.47 (0.44→0.50) | P = 0.001 | 0.05→0.14 | 0.002 | 0.003 | 0.22 |
| PIF | 0.66 (0.61→0.71) | 0.55 (0.48→0.56) | 0.68 (0.63→0.71) | P = 0.83 | –0.13→0.16 | 0.03 | 0.96 | 0.05 |

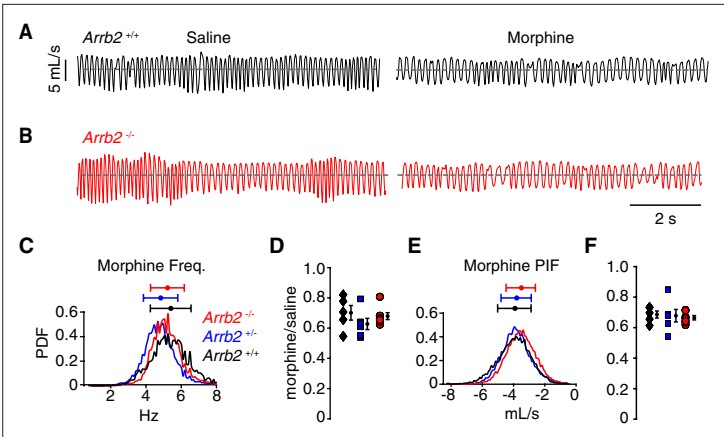

**Figure 3.** Basal respiration and OIRD in *Arrb2* littermates in hypercapnic conditions. (**A**) Example breathing trace in hypercapnic conditions (21% $O_2$, 5% $CO_2$) following IP saline (left) and morphine (right) for *Arrb2+/+*. (**B**), Example breathing traces from *Arrb2-/-*. (**C**), Histogram of instantaneous respiratory frequency (Hz) for all breaths in morphine from *Arrb2-/-* (red, combined from n = 7 animals), *Arrb2+/-* (blue, n = 6), and *Arrb2+/+* (black, n = 5). PDF, probability density function. Top, mean (circle)± standard deviation (bars). Respiratory measurements for saline and morphine are reported in Table 3. (**D**), OIRD defined as the ratio of average respiratory frequency in morphine to saline for *Arrb2-/-*, *Arrb2+/-*, and *Arrb2+/+*. Mean (circle)± standard deviation (bars). Data included in Table 4. (**E–F**), Analysis of peak inspiratory airflow (PIF) as in **C–D**. but for peak inspiratory airflow. Single and Two Factor ANOVA and unpaired t-test statistics reported in Table 4. Statistics were not corrected for multiple comparisons to maximize the possibility of identifying differences between *Arrb2* genotypes.

The online version of this article includes the following figure supplement(s) for figure 3:

**Source data 1.** Raw respiratory data, OIRD ratio, and statistical tests for recordings performed in hypercapnic conditions.

**Figure supplement 1.** Power analysis to determine the OIRD effect size given the cohort sizes and data in normoxic and hypercapnic experimental conditions.

(*Figure 4C–D*), although this likely stems from the differences in cohort sizes (6 vs 20). In conclusion, a MOR agonist that substantially activates ß-arrestin 2 signaling similarly slows the preBötC rhythm in mice lacking the *Arrb2* gene and the littermate controls.

## Discussion

The proposed unique importance of MOR-dependent ß-arrestin 2 signaling in OIRD has motivated the development of biased agonists for analgesia. However, the recent failure to reproduce this result has called model into question (*Kliewer et al., 2020*). Therefore, the goal of our studies was to test the null hypothesis that germline deletion of *Arrb2* does not attenuate OIRD. The results from our in vivo studies under normoxic and hypercapnic conditions, as well as our in vitro studies, failed to reject this null hypothesis. In order to directly compare to previous results, we sufficiently powered our cohort size to identify effect sizes reported in *Raehal et al., 2005* (~50% less OIRD), and the statistical

**Table 3.** Mean and confidence interval for hypercapnic condition raw respiratory frequency and peak inspiratory airflow after saline and morphine intraperitoneal injection.

|  | *Arrb2 -/- mean ± SEM* | *Arrb2 -/- 95% CI* | *Arrb2+/- mean ± SEM* | *Arrb2 +/- 95% CI* | *Arrb2 +/+ mean ± SEM* | *Arrb2 +/+ 95% CI* |
|---|---|---|---|---|---|---|
| Freq. saline (Hz) | 7.68 ± 0.19 | 7.31 → 8.05 | 7.69 ± 0.16 | 7.38 → 8.00 | 7.77 ± 0.39 | 7.01 → 8.53 |
| Freq. morphine | 5.20 ± 0.10 | 5.00 → 5.40 | 4.79 ± 0.19 | 4.42 → 5.16 | 5.38 ± 0.18 | 5.02 → 5.73 |
| PIF saline (mL/s) | −5.34 ± 0.27 | −4.81 → −5.87 | −5.75 ± 0.25 | −5.26 → −6.24 | −5.71 ± 0.20 | −5.32 → −6.10 |
| PIF morphine | −3.54 ± 0.16 | −3.22 → −3.85 | −3.86 ± 0.15 | −3.57 → −4.15 | −3.93 ± 0.24 | −3.46 → −4.40 |

**Table 4.** OIRD values of respiratory frequency and peak inspiratory airflow in hypercapnic conditions and the several types of statistical tests.

| | Arrb2 -/- OIRD median or mean (95% CI) | Arrb2+/- mean (95% CI) | Arrb2 +/+ mean (95% CI) | Arrb2 -/- vs. Arrb2 +/+ (Mann-Whitney or unpaired t-test, two tail) | Arrb2 -/- vs. Arrb2 +/+ (95% CI) | Kruskal-Wallis or One-way anova | Two-way Anova - regression (interaction) |
|---|---|---|---|---|---|---|---|
| Freq. | 0.66 | 0.63 (0.55→0.71) | 0.70 (0.60→0.80) | P = 0.52 (MW) | −0.11→0.15 | 0.16 (KW) | 0.45 |
| PIF | 0.66 (0.64→0.68) | 0.68 (0.61→0.77) | 0.69 (0.64→0.72) | P = 0.46 (t) | −0.04→0.08 | 0.86 (anova) | 0.96 |

tests were performed conservatively by not correcting for multiple comparisons. Combined, these three independent assays demonstrate that the germline knockout of *Arrb2* does not attenuate OIRD.

Unlike other studies, we designed ours with five important features to ensure a robust conclusion. First, the OIRD comparisons were made between littermates in an effort to control for any strain specific effects on breathing and opioid sensitivity (*Bubier et al., 2020*). Second, OIRD was defined as the comparison of breathing after IP injection of saline and morphine in the same animal to control for any within-animal specific breathing variation. Third, we analyzed the multiple breathing parameters with and without averaging across long stretches of breathing. Four, OIRD was measured under a hypercapnic state for a more precise quantification. And fifth, we validated our in vivo studies by directly measuring the impact of a MOR ligand on the preBötC rhythm, the key site for OIRD. Future studies should also confirm an unchanged sensitivity to opioids in other brain or peripheral sites that contribute to OIRD.

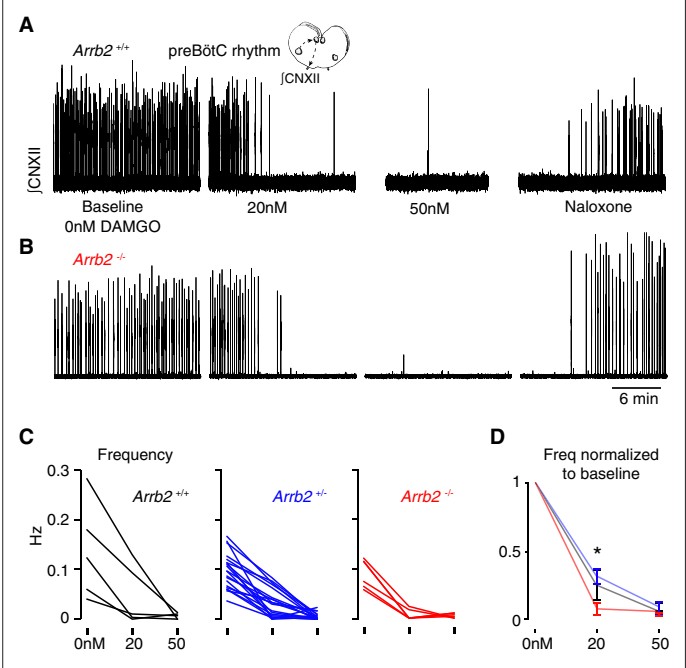

**Figure 4.** Slowing of preBötC rhythmicity with a MOR agonist from *Arrb2* littermates. A, Neonatal medullary preBötC slice preparation. PreBötC inspiratory activity is measured via the hypoglossal nerve rootlet (∫CNXII, arbitrary units). Example twenty minute recordings from an *Arrb2*+/+ slice where the rhythm is recorded at baseline (0 nM DAMGO), then 20 nM and 50 nM DAMGO, and then with the addition of 100 nM Naloxone. (**B**), As in **A**, but example recording from an *Arrb2*-/- littermate. (**C**), The preBötC frequency (Hz) for each slice from *Arrb2*+/+ (black, n = 5), +/- (blue, n = 20), and -/- (red, n = 6) littermates at each DAMGO dose. Average frequency (bursts per second) is measured during the last five minutes of each recording. (**D**), Mean ± SEM for normalized frequency for each genotype at each DAMGO dose. *, p-value < 0.05 for Kruskal-Wallis test. p-Values for pairwise Mann-Whitney: 0.10 for +/+ vs -/-, 0.45 for +/+ vs +/-, 0.02 for±vs -/-.

The online version of this article includes the following figure supplement(s) for figure 4:

**Source data 1.** Raw data and statistical tests for in vitro OIRD recordings.

One difference between our study and the original *Arrb2* study by Raehal et al. was our method of morphine delivery. Raehal et al. reported that at the maximal concentration of morphine delivered (150 mg/kg subcutaneous), the respiratory rate in *Arrb2* knockouts was depressed to half that of wild-type controls (~20% vs 40% depression of breathing rate). In our study, although we delivered 20 mg/kg morphine by IP injection, we observed an even larger depression of breathing in all three *Arrb2* genotypes (60% in normoxic and 30% in hypercapnic conditions), indicating our assay was sufficient to observe the reported attenuated OIRD response in *Arrb2-/-* mice. Beyond this, several important future studies to exhaust other methodological limitations include the measurement of respiratory depression in *Arrb2-/-* and littermates with other abused opioids like oxycodone and heroin, to temporally or spatially delete *Arrb2* using *Arrb2*^flox/flox alleles to overcome possible compensatory mechanisms, and the study of biased agonists in models where potent opioids like fentanyl are lethal. Additional or other mechanisms may underlie the fatal apnea that was not studied here.

The core premise for the development of MOR biased agonists is that *Arrb2*-dependent signaling provides a molecular mechanism to dissociate analgesia from respiratory depression. Our findings do not support this claim. Consistently, a recent study demonstrated that an opioid receptor ligand that does not induce MOR ß-arrestin 2-dependent signaling still temporarily induces OIRD (*Uprety et al., 2021*). Given all of this, how then do some biased agonists show analgesia while minimizing respiratory depression? Perhaps biased agonists are just partial MOR agonists for activation of G-protein signaling. In this case, we imagine analgesia is more sensitive than respiratory depression, and therefore certain concentrations of MOR ligand enable these two effects to be separated. This would be akin to providing a lower dose of standard opioid-like drugs. Regardless, our results, along with similar data from another recent study (*Kliewer et al., 2020*), refute the foundational model that *Arrb2* selectively mediates OIRD and suggest that now we must reconsider and in the future reinvestigate the mechanism of biased agonism in vivo.

# Materials and methods

**Key resources table**

| Reagent type (species) or resource | Designation | Source or reference | Identifiers | Additional information |
|---|---|---|---|---|
| Strain, strain background (*Mus musculus*, male and female) | *Arrb2-/-* | The Jackson Laboratory | 011130 | |
| Strain, strain background (*Mus musculus*, male and female) | C57Bl/6 J | The Jackson Laboratory | 000664 | |
| Peptide, recombinant protein | DAMGO | Abcam | Ab12067 | |
| Chemical compound, drug | Morphine sulfate | Henry Schein | 057202 | |
| Chemical compound, drug | Naloxone | Sigma Aldrich | N7758 | |
| Software, algorithm | Matlab | Mathworks | | https://github.com/YackleLab/Opioids-depress-breathing-through-two-small-brainstem-sites |

## Animals

*Arrb2 -/-* mice (*Bohn et al., 1999*) were bred to C57BL/6 to generate heterozygous F1. The F1 littermates were then crossed to make *Arrb2-/-*, *Arrb2-/+*, and *Arrb2+/+* (F2). Mice were housed in a 12 hour light/dark cycle with unrestricted food and water. Mice were given anonymized identities for experimentation and data collection. All animal experiments were performed in accordance with national and institutional guidelines with standard precautions to minimize animal stress and the number of animals used in each experiment. Institutional Animal Care and Use Committee approval number AN181239.

## Plethysmography and respiratory analysis

Plethysmography and respiratory analysis were performed as in *Bachmutsky et al., 2020*. Briefly, on the first recording day, adult (6–12 weeks) *Arrb2-/-*, *Arrb2-/+*, and *Arrb2+/+* mice were administered

IP 100 μL of saline and placed in an isolated recovery cage for 15 min. After, individual mice were then monitored in a 450 mL whole animal plethysmography chamber at room temperature (22°C) in 21% $O_2$ balanced with $N_2$ (normoxic condition) or 21% $O_2$, 5% $CO_2$ balanced with $N_2$ (hypercapnic condition). After 1 day, the same protocol was used to monitor breathing after IP injection of morphine (20 mg/kg, Henry Schein 057202). The morphine recordings under normoxic and hypercapnic conditions were separated from saline recordings by at least 3 days. Each breath was automatically segmented based on airflow crossing zero as well as quality control metrics. Respiratory parameters (e.g. peak inspiratory flow, instantaneous frequency) for each breath, as well as averages, were then calculated. Reported airflow in mL/sec. is an approximate of true volumes. The analysis was performed with custom Matlab code available on Github with a sample dataset (https://github.com/YackleLab/Opioids-depress-breathing-through-two-small-brainstem-sites). All animals in the study were included in the analysis and cohorts including all these genotypes were run together.

## Statistics

A power analysis was performed using the reported effect size from Raehal et al. In this case, 1–4 mice were necessary to observe a statistically significant result. Each cohort (*Arrb2*+/+, +/-, -/-) exceeded 4. Statistical tests were performed on the ratio of IP morphine to IP saline for instantaneous respiratory frequency and peak inspiratory flow separately for normoxic and hypercapnic conditions. A Shapiro Wilks test was first done to determine if the data was normally distributed (*Figure 2—source data 1*, *Figure 3—source data 1*). If normal, a single factor ANOVA was performed to determine any differences among the three genotypes (alpha <0.05). In the instance the p-value was <0.05, the Tukey HSD post-hoc test was done to determine which of the pairwise comparisons were statistically different (alpha <0.05). Additionally, one-way unpaired parametric T-tests were used to compare *Arrb2* +/+ and -/- genotypes (alpha <0.05). If the data failed to pass the Shapiro Wilks test, then the non-parametric Kruskal-Wallis test was used to determine if any differences (alpha <0.05). And the Mann-Whitney U test was used to compare *Arrb2*+/+ and -/- genotypes (alpha <0.05). A two-way ANOVA with regression was used to determine interactions between each of the genotypes IP saline and IP morphine values. To determine the power of our data, we compared hypothetical *Arrb2*+/+ and measured *Arrb2*-/- means. The power calculation included our cohort size and the measured standard deviation of the two genotypes. A similar statistical approach was used to analyze the in vitro data (*Figure 4—source data 1*). All the above statistics were performed using the publicly available Excel package 'Real Statistics Functions' SPSS and Matlab.

## Slice electrophysiology

Rhythmic 550–650 μm-thick transverse medullary slices which contain the preBötC and cranial nerve XII (XIIn) from neonatal *Arrb2* -/-, +/-, +/+ mice (P0-5) were prepared as described (*Bachmutsky et al., 2020*). Slices were cut in ACSF containing (in mM): 124 NaCl, 3 KCl, 1.5 $CaCl_2$, 1 $MgSO_4$, 25 $NaHCO_3$, 0.5 $NaH_2PO_4$, and 30 D-glucose, equilibrated with 95% $O_2$ and 5% $CO_2$ (4 °C, pH = 7.4). Recordings were performed in 9 mM at a temperature of 27°C. Slices equilibrated for 20 min before experiments were started. The preBötC neural activity was recorded from CNXII rootlet. Activity was recorded with a MultiClamp700A or B using pClamp9 at 10,000 Hz and low/high pass filtered at 3/400 Hz. After equilibration, baseline activity and then increasing concentrations of DAMGO (ab120674) were bath applied (20 nM, 50 nM). After the rhythm was eliminated, 100 nM Naloxone (Sigma Aldrich N7758) was bath applied to demonstrate slice viability. The rate was determined from the last 5 min of each 20-min recording and rhythmic activity was normalized to the first control recording for dose response curves.

## Additional information

### Funding

| Funder | Grant reference number | Author |
|---|---|---|
| NIH Office of the Director | DP5-OD023116 | Kevin Yackle |

| Funder | Grant reference number | Author |
| --- | --- | --- |
| Program for Breakthrough Biomedical Research | | Kevin Yackle |

The funders had no role in study design, data collection and interpretation, or the decision to submit the work for publication.

## Author contributions

Iris Bachmutsky, conceptualization, data-curation, formal-analysis, i.b.-and-a.d.-conducted-all-in-vivo-experiments.-p.w.-and-i.b.-conducted-in-vitro-experiments., investigation, methodology, software, validation, visualization, writing-original-draft, writing-review-and-editing; Xin Paul Wei, conceptualization, data-curation, investigation, methodology, p.w.-and-i.b.-conducted-in-vitro-experiments.-i.b.-analyzed-all-data., validation, writing-review-and-editing; Adelae Durand, data-curation, investigation, writing-review-and-editing; Kevin Yackle, conceptualization, funding-acquisition, project-administration, supervision, visualization, writing-original-draft, writing-review-and-editing

## Author ORCIDs

Xin Paul Wei (iD) http://orcid.org/0000-0002-6621-3143
Kevin Yackle (iD) http://orcid.org/0000-0003-1870-2759

## Ethics

All animal experiments were performed in accordance with national and institutional guidelines with standard precautions to minimize animal stress and the number of animals used in each experiment. All animal protocols have been approved by the UCSF 'Office of Research'.approval number AN181239.

## Decision letter and Author response

Decision letter https://doi.org/10.7554/eLife.62552.sa1
Author response https://doi.org/10.7554/eLife.62552.sa2

---

# Additional files

## Supplementary files

• Transparent reporting form

## Data availability

The data generated in Figures 2-4 are provided in the source files.

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
