## [Decision Letter]

**Acceptance summary:**

There is definitely great interest in the mechanisms that underlie the adverse side effects that occur when opioids are used to relieve pain. Your study has very convincingly demonstrated that a prevailing hypothesis, namely that Mu opioid receptor-induced ß-arrestin 2-dependent signaling is responsible for opioid respiratory depression, cannot be supported. Alternative mechanisms must be investigated.

**Decision letter after peer review:**

Thank you for submitting your article "ß-arrestin 2 germline deletion does not attenuate opioid respiratory depression" for consideration by *eLife*. Your article has been reviewed by 3 peer reviewers, and the evaluation has been overseen by a Reviewing Editor and Christian Büchel as the Senior Editor. The following individuals involved in review of your submission have agreed to reveal their identity: Jack L Feldman (Reviewer #1); Bryan Roth (Reviewer #3).

The reviewers have discussed the reviews with one another and the Reviewing Editor has drafted this decision to help you prepare a revised submission.

Essential revisions:

There is considerable support for your manuscript. We appreciate that your submission was intended as a Short Report, however, there is consensus among the reviewers that were you to provide the new experiments indicated, *eLife* should definitely consider publishing as a regular manuscript. We hope that you are able to complete these experiments as they will greatly increase the impact of your manuscript. Key points of the Reviewers comments are noted below.

Since the in vivo experiments were germline knockouts, some comment should be made as to the possibility that the results are impacted by effects in sites other than the preBotizinger Complex or the parabrachial nuclear complex. The in vitro experiments address this implicitly, but an explicit sentence or two would be helpful.

Deaths in OIRD are typically the result of asphyxiation, the result of a long lasting apnea. In this study, as well as all of their references, the depression of breathing in mice was far from apnea. How does this impact the significance of these findings to OIRD in humans?

One methodological concern is the use of morphine as the µOR agonist. Most of the prescribed and/or abused µOR agonists (e.g., oxycodone, fentanyl, heroin) are much more potent than morphine (https://www.ncbi.nlm.nih.gov/books/NBK537482/table/appannex6.tab2/ ), and account for most of the OIRD deaths. While not necessary for this paper, the authors might consider using more potent agonists in follow up studies.

One reviewer suggests that the writing could be a bit more precise in distinguishing between gas mixtures and presumptive blood gas concentrations of CO2 or O2. For example, Line 79: "hypercapnia" is not a "gas". Do you mean "hypocapnic gas mixture"? Note: same issue elsewhere with using "normoxia" to refer to a different gas mixture.

Statistics: The proper way to analyze the data is a two-way repeated measures ANOVA with the factors genotype and treatment.

The authors need to define the β-error and confidence intervals to state that there was no difference between genotypes (within given confidence intervals).

The DAMGO slice experiments refer to a "historical control". In order to be conclusive, the authors must repeat these experiments in wild-type mice (in a blinded manner!) and do a proper side-by-side comparison.

In order to convincingly contradict previous results, the experimental design should be comprehensive. In this specific case, testing more than a single dose of morphine would make the results more convincing.

One reviewer believes that the electrophysiological studies are incomplete. At a minimum they should provide dose-response findings in WT and KO slices to an endogenous peptide and morphine. Additionally, they would need to provide some estimates of experimental variability and, at the least, biological and technical replicates. Finally, they would need to provide statistical analysis of the findings. Simply exposing N=1 slice to DAMGO is not persuasive.

The authors may wish to consider that since these mice have had bArr2 deleted from conception, if this molecule is involved in the actions of opioids it is possible that compensatory changes have occurred during development to abrogate the effect in adults. Of course, this does not explain the difference between these results and prior results but does provide additional context for the findings.

---

## [Author Response]

Essential revisions:There is considerable support for your manuscript. We appreciate that your submission was intended as a Short Report, however, there is consensus among the reviewers that were you to provide the new experiments indicated, eLife should definitely consider publishing as a regular manuscript. We hope that you are able to complete these experiments as they will greatly increase the impact of your manuscript. Key points of the Reviewers comments are noted below.Since the in vivo experiments were germline knockouts, some comment should be made as to the possibility that the results are impacted by effects in sites other than the preBotizinger Complex or the parabrachial nuclear complex. The in vitro experiments address this implicitly, but an explicit sentence or two would be helpful.

Our study is inconsistent with the original description of OIRD in Arrb2 germline knockouts (Raehal et al. 2005). in vitro, we demonstrate the preBötC is similarly impacted by opioids in all three genotypes. However, as the reviewer notes, the similar impact upon the preBötC may only partially explain the in vivo results. We now note on Page 8, line 156-157 that future studies should explore the sensitivity of other important central and peripheral OIRD sites in all three genotypes.

Deaths in OIRD are typically the result of asphyxiation, the result of a long lasting apnea. In this study, as well as all of their references, the depression of breathing in mice was far from apnea. How does this impact the significance of these findings to OIRD in humans?

The focus of our work was to reproduce the results from Raehal et al. 2005 which led to the conclusion that Arrb2 dependent signaling is a key mediator of OIRD. This provided the rationale for developing MOR ‘biased agonists’. Despite not assaying lethality or other abused opioids, our preclincal results remain significant as they inform the groups and companies currently developing ‘biased agonists’ as well as the NIH programs currently funding or planning to fund such approaches. We have added such comments into our discussion on Page 8, lines 165-167.

One methodological concern is the use of morphine as the µOR agonist. Most of the prescribed and/or abused µOR agonists (e.g., oxycodone, fentanyl, heroin) are much more potent than morphine (https://www.ncbi.nlm.nih.gov/books/NBK537482/table/appannex6.tab2/ ), and account for most of the OIRD deaths. While not necessary for this paper, the authors might consider using more potent agonists in follow up studies.

This is an important point and future experiment that must be conducted. We had added this comment into our discussion on Page 8, lines 165-167.

One reviewer suggests that the writing could be a bit more precise in distinguishing between gas mixtures and presumptive blood gas concentrations of CO2 or O2. For example, Line 79: "hypercapnia" is not a "gas". Do you mean "hypocapnic gas mixture"? Note: same issue elsewhere with using "normoxia" to refer to a different gas mixture.

Thank you for this feedback. We have corrected our terminology throughout the text.

Statistics: The proper way to analyze the data is a two-way repeated measures ANOVA with the factors genotype and treatment.

The results of a Two-Way ANOVA with regression are now included in Table 2 and 4. A regression was used since the number of mice for each Arrb2 genotype is unbalanced. The only interaction observed was the PIF in the normoxic condition. Upon visualization of the means, it appears the interaction is due to the increased saline PIF in the Arrb2+/- mice.

**Author response image 1. sa2fig1:** 

The authors need to define the β-error and confidence intervals to state that there was no difference between genotypes (within given confidence intervals).

The confidence intervals for the respiratory parameters, OIRD, and t-test/Mann-Whitney U are now included in Tables 1-4. We now comprehensively describe the power of our study in Figure 3-Supplement 1 and text Page 6, Lines 111-121.

The DAMGO slice experiments refer to a "historical control". In order to be conclusive, the authors must repeat these experiments in wild-type mice (in a blinded manner!) and do a proper side-by-side comparison.

This experiment has been conducted in litters containing Arrb2+/+, +/-, and -/- littermates in a blinded fashion. The data is displayed in Figure 4 and described on Page 6-7, Lines 121-135.

In order to convincingly contradict previous results, the experimental design should be comprehensive. In this specific case, testing more than a single dose of morphine would make the results more convincing.We agree with the reviewer that a dose response curve would more comprehensively describe OIRD. However, the single dose used produced a profound respiratory depression equivalent to or even larger than what is displayed in Figure 5 of Raehal et al. 2005. Given this, we believe the single chosen dose is sufficient to refute the prior claim, and this is the primary message of the manuscript.Please note the impact of COVID-19 on our revision experiments. The shutdown necessitated a culling of our mouse colony. This required us to re-breed all F1 and then F2 animals from the very few Arrb2-/- we retained. This breeding alone has taken more than 8 months. Furthermore, the Arrb2-/- stocks at Jax were culled and these mice are backordered due to breeding to re-establish the colony. So far, the F2 mice we have been generated have been dedicated to our in vitro studies.One reviewer believes that the electrophysiological studies are incomplete. At a minimum they should provide dose-response findings in WT and KO slices to an endogenous peptide and morphine. Additionally, they would need to provide some estimates of experimental variability and, at the least, biological and technical replicates. Finally, they would need to provide statistical analysis of the findings. Simply exposing N=1 slice to DAMGO is not persuasive.

As described in Point 7, we have repeated our in vitro study with a dose response curve in all three Arrb2 genotypes (Figure 4). We now display the variability of our data and performed the appropriate statistical tests.

We chose to conduct these in vitro studies with DAMGO since this peptide is the archetype MOR agonist used in biochemical studies to demonstrate if MOR ligands have signaling bias (for example, see Schmid et al. 2017). Although the pharmacokinetics of morphine make it more suitable for the in vivo studies, DAMGO is the most appropriate MOR ligand to investigate activation of both G_α_i and ß-arrestin 2 dependent MOR signaling in vitro.

The authors may wish to consider that since these mice have had bArr2 deleted from conception, if this molecule is involved in the actions of opioids it is possible that compensatory changes have occurred during development to abrogate the effect in adults. Of course, this does not explain the difference between these results and prior results but does provide additional context for the findings.

As described in Point 7, we have repeated our in vitro study with a dose response curve in all three Arrb2 genotypes (Figure 4). We now display the variability of our data and performed the appropriate statistical tests.

We chose to conduct these in vitro studies with DAMGO since this peptide is the archetype MOR agonist used in biochemical studies to demonstrate if MOR ligands have signaling bias (for example, see Schmid et al. 2017). Although the pharmacokinetics of morphine make it more suitable for the in vivo studies, DAMGO is the most appropriate MOR ligand to investigate activation of both G_α_i and ß-arrestin 2 dependent MOR signaling in vitro.